# A Rapid and Sensitive Method for the Simultaneous Determination of Multipolar Compounds in Plant Tea by Supercritical Fluid Chromatography Coupled to Ion Mobility Quadrupole Time-of-Flight Mass Spectrometry

**DOI:** 10.3390/foods11010111

**Published:** 2022-01-01

**Authors:** Zi-Xuan Yue, Jun Cao

**Affiliations:** 1College of pharmacy, Hangzhou Normal University, Hangzhou 311121, China; 2020112012153@stu.hznu.edu.cn; 2College of Material Chemistry and Chemical Engineering, Hangzhou Normal University, Hangzhou 311121, China

**Keywords:** supercritical fluid chromatography, caulis dendrobii, matrix solid phase dispersion, ion mobility quadrupole time-of-flight mass spectrometry, multipolar compounds, plant tea

## Abstract

In this study, matrix solid phase dispersion (MSPD) microextraction combined with supercritical fluid chromatography-ion mobility quadrupole time-of-flight mass spectrometry (SFC/IM-QTOF-MS) was used to analyze the multipolar compounds in plant tea. The parameters of stationary phase, mobile phase, make-up solution, temperature, and back pressure were optimized. The target analytes were gradient eluted in 8 min by supercritical CO_2_ on a Zorbax RX-SIL column. Collisional Cross Section (CCS) values for single and multiple fields were measured. A series of validation studies were carried out under the optimal conditions, and the linear relationship and reproducibility were good. The limits of detection were 1.4 (Scoparone (1))~70 (Naringenin (4)) ng/mL, and the limits of quantification were 4.7 (Scoparone (1))~241 (Naringenin (4)) ng/mL. The recoveries of most compounds ranged from 60.7% to 127%. As a consequence, the proposed method was used for the separation and quantitative analysis of active ingredients in caulis dendrobii.

## 1. Introduction

Supercritical fluid chromatography (SFC) is generally is a chromatography technique in which the mobile phase is supercritical carbon dioxide (SCCO_2_) or subcritical carbon dioxide [1]. These fluids are close to or surpass critical temperature and/or pressure of CO_2_, and have the advantage of low viscosity and high diffusivity, thus, enabling rapid analysis of target compounds with high resolution [2]. In addition, CO_2_ is easily regenerated and has lower toxicity than conventional chromatographic solvents, thereby, providing a “more environmentally friendly” analytical method [3]. SFC can be used for chiral applications, especially on a preparative scale in the pharmaceutical industry [4]. At the same time, SFC is also used for the analysis of active ingredients in natural products [5]. Yang et al. achieved baseline separation of 12 flavonoids on a ZORBAX RX-SIL column using SFC-DAD [6]. Gao et al. isolated six flavonoid aglycones by using Poroshell 120 EC-CN as a chromatographic column and adding 20 mM [Bmim] [BF4] as a mobile phase modifier in SFC [7]. However, most of these studies only covered flavonoids widely present in traditional Chinese medicine and could not analyze compounds with different structures at the same time. As a consequence, it makes sense to further expand the application of SFC.

Ion mobility quadrupole time-of-flight mass spectrometry (IM-QTOF-MS) is a new 2D mass spectrometry technique combining ion mobility separation and mass spectrometry [8]. The principle is that ions can be separated according to their size and shape based on the different cross-sections when they collide with buffer gas in a drift tube [9]. IM-QTOF-MS combines the characteristics of ion mobility technology, which is sensitive, fast, and can provide ionic structure information [10]. It is showing more and more advantages in the analysis of chemical isomers, the interaction analysis of biological macromolecules, and so on [11,12,13]. Ben-Nissan et al. discussed the application of IM-MS in the study of the structure/function of protein complexes [14]. Masike et al. studied the application of IM-MS in natural products [15]. On the other hand, supercritical fluid chromatography tandem mass spectrometry (SFC-MS/MS) has attracted more and more attention due to its advantages of higher efficiency, shorter analysis time, and accurate characterization of compounds [16]. Seiwert et al. combined the ACQUITY UPC^2^ system with SYNAPT GS2 Q-TOF for the determination of trace organic chemicals and ozonation products of metformin in municipal wastewater [17]. Jiang et al. used the Nexera UC SFC system in tandem with the LC/MS triple quadrupole mass spectrometer for rapid determination of alkyl phenols and ethoxy compounds in leaf vegetables [18]. However, there are few studies on supercritical fluid chromatography coupled with ion mobility quadrupole time-of-flight mass spectrometry (SFC/IM-QTOF-MS). Therefore, it is imperative and important to further deepen the research of SFC/IM-QTOF-MS.

Caulis dendrobii is a perennial herb of Dendrobium genus in Orchidaceae. There are many types of caulis dendrobii found in nature, mainly including dendrobium nobile, dendrobium chrysotoxum, and dendrobium candidum [19]. Modern clinical studies have shown that caulis dendrobii has great antioxidant, antitumor, and antifatigue activities and is effective in lowering uric acid, hypoglycemia, and immunoregulation [20]. Caulis dendrobii is rich in medicinal active ingredients, mainly including polysaccharides, flavonoids, alkaloids, astragalus, amino acids, phenanthines, dibenzyl, and phenols [21]. For example, flavonoids have a wide range of biochemical and pharmacological properties and have been used for the purposes of anti-inflammatory, immunosuppressive, cytotoxic, etc. [22]. Considering the important functions of these active ingredients, it is very valuable to study the active substances in caulis dendrobii. Yang et al. used high performance liquid chromatography (HPLC) to simultaneously determine phenolic compounds (biphenyl, phenanthrene, and fluorenone) in caulis dendrobii [23]. Zhang et al. used high performance liquid chromatography coupled to mass spectrometry (HPLC/MS) to analyze trans and cis isomers of 2-glucoxy cinnamic acid and derivatives of coumarin in caulis dendrobii [24]. However, traditional methods for analysis of caulis dendrobii have many limitations, such as long extraction time, serious environmental pollution, and the inability to enrich. Therefore, it is necessary to develop a simple, rapid, and environmentally friendly analysis method.

In this study, a rapid MSPD combined with SFC/IM-QTOF-MS method was developed for the isolation of multiple types of active ingredients from caulis dendrobii. In order to optimize the separation performance, various parameters affecting the experiment, including the type of stationary phase, organic modifier, the content of acid additive, pressure, and temperature were studied. In addition, selectivity, resolution, and analysis time were optimized using standard materials as test samples. To determine the target compound more accurately, we measured CCS values for single and multiple fields. Finally, the established method was validated and applied to the analysis of active ingredients in caulis dendrobii, which were Scoparone (1), Dendrobine (2), Dendrophenol (5), Luteolin (6), Naringenin (4), Apigenin (3).

## 2. Materials and Methods

### 2.1. Chemicals and Materials

Carbon dioxide (CO_2_) (purity 99.5%) was purchased from Yinglai gas company (Guangzhou, China). HPLC grade methanol was obtained from Tedia Company Inc. (Fairfield, OH, USA). Formic acid was acquired from Alfa Aesar Chemical Co., Ltd. (Tianjin, China). Acetic acid was bought from Lingfeng Chemicals Co., Ltd. (Hangzhou, China). Trifluoroacetic acid (TFA), ammonium formate, and ammonium acetate were provided by Sigma Aldrich Trading Co., Ltd. (Shanghai, China). Seven reference materials including Scoparone (1), Dendrobine (2), (5), Luteolin (6), Naringenin (4), and Apigenin (3) and (7) were supplied by Shanghai fusion Biotechnology Co., Ltd. (Shanghai, China). Their structures are displayed in Figure 1. Caulis dendrobii was from the Yandangshan area (Wenzhou, China). All water used in this experiment was highly purified water provided by Merck Chemical Technology Co., Ltd. (Shanghai, China).

### 2.2. Instruments

This experiment was carried out on an Agilent SFC 1260 Infinity series Agilent 6560 IM-QTOF-MS (Santa Clara, CA, USA, Agilent Technologies) analysis system. The SFC system consists of an SFC binary pump, 1260 Infinity LC system, degasser, autosampler, column thermostat, back pressure regulator (BPR), and DAD detector. Agilent Open Lab Chem Station Edition C.01.05 software was used for instrument control and data collection. The following columns were used in this study: ZORBAX RX-SIL column (150 mm × 4.6 mm, 5 μm), RP Amide column (250 mm × 4.6 mm, 5 μm), Agilent NH_2_ column (150 mm × 4.6 mm, 5 μm), and an Agilent SB-CN column (150 mm × 4.6 mm, 5 μm). The optimized BPR back pressure and temperature were set to 60 °C and 100 bar respectively, and the DAD detection wavelength was set to 280 nm. CO_2_ was used as mobile phase component A and methanol (containing 0.1% TFA and 8% H_2_O) was used as mobile phase B. The elution gradient conditions were as follows: 0 min~15%B, 2 min~18%B, 4 min~20%B, 7 min~35%B, and 8 min~35%B. The injection volume, flow rate, and column temperature were 5 μL, 3.0 mL/min, and 25 °C, respectively.

Mass spectrometry used the ESI source to perform a full scan in the positive ion mode, and the scan range was from 100 to 400. The mass spectrometer parameters were as follows: drying gas temperature 350 °C, drying gas speed 12 L/min, capillary voltage set to 3500 V, Skimmer voltage maintained at 65 V, oct 1 RF Vpp voltage was 750 V, atomizer gas pressure was 35 psi, and the fragmentor voltage was 175 V.

### 2.3. Matrix Solid-Phase Dispersion Process

Different types of caulis dendrobii samples were dried in an oven (2 h at 50 °C), crushed into powder, and passed through a 40-mesh sieve. Then, 30 mg of the caulis dendrobii sample was accurately weighed and moved into agate mortar, 30 mg of C18 dispersant was added, and the grinding was carried out at a constant speed for 1 min. The mixed sample was transferred to a 1 mL solid phase extraction column, and the column was compacted up and down with sieve plates. Finally, 200 μL methanol was used for elution, and the eluate was collected in a 1.5 mL centrifuge tube. Before injection into the SFC-IM-QTOF-MS analysis, the eluate was centrifuged at a high speed at 13,000 rpm for 5 min. For sample treatment, we referred to the literature and Chinese pharmacopoeia [25,26].

## 3. Results and Discussions

### 3.1. The Selection of Stationary Phase

The selection of an appropriate stationary phase played a crucial role in the separation of compounds with an eye to the extraordinary needs of SFC. In order to separate seven different target analytes, four types of stationary phases were tested, such as ZORBAX RX-SIL, RP Amide, Agilent NH_2_, and Agilent SB-CN. BPR back pressure and temperature were set at 60 °C and 100 bar, respectively, and the DAD detection wavelength was set at 280 nm. CO_2_ was used in mobile phase A, methanol (containing 0.1% TFA and 8% H_2_O) was used in mobile phase B, and 0.1% acetic acid was used in the make-up solution. As can been seen from Figure 2 and Figure 3, the separation effect of RP Amide was the worst among these stationary phases. In the total ion flow diagram, there was no obvious peak shape on the column of RP Amide. The reason for the short retention time may be that the polarity of the stationary phase was too large, which resulted in the co-elution of the tested compound. However, the NH_2_ column had certain selectivity to the compounds, and naringin (4) and Luteolin (6) were not retained in the column. The stationary phase of NH_2_ was different from the others in that it could provide an additional hydrophilic effect for positive phase analysis; thus, the retention orders of analytes were slightly different. The best performance for the stationary phase of SFC was RX-SIL packed with 150 mm × 4.6 mm containing 5 µm particles. Moreover, the resolutions obtained by using RX-SIL were greater than 1.5, because RX-SIL mainly have been shown to interact with target analytes via a combination of hydrogen bonding and dipole-dipole interactions. In addition, the separation effect of SB-CN was slightly worse than RX-SIL. Compounds concentrated in the first three minutes of peak, the separation degree was less than RX-SIL column, as the π electron interaction of cyano group could lead to the different selectivity to tested compounds. Therefore, the stationary phase of RX-SIL was selected for further experiments.

### 3.2. The Selection of Mobile Phase

A modifier, which was usually added to CO_2_, was necessary to improve separation selectivity, peak shape, and analyte solubility. However, the modifier had several side-effects, such as increasing mobile phase density, adjusting selectivity, and so on. According to the reported literature and the results of the pre-experiment, methanol was commonly used as a modifier of SFC, which was greatly miscible with CO_2_. The target compounds contained hydroxyl and phenolic hydroxyl groups, which could interact with the silicon hydroxyl groups existing in the stationary phase, such as a hydrogen bond, dipole–dipole interactions, and strong adsorption, resulting in the tailing phenomenon. Therefore, additives were needed and optimized to improve the chromatographic separation. TFA, formic acid, and acetic acid were selected as additives. The results are shown in Figure 4a. The analysis response and peak shape of TFA were better than the other two acids, the lowest response was naringenin (4) with a response value of 0.9 × 10^6^. No dendrobine (2) was detected when formic acid and acetic acid were used as additives. The addition of TFA in the mobile phase could inhibit the dissociation of the compound and the silicon hydroxyl group so as to make the solution neutral. Thus, TFA could influence the interaction between the target analytes and the silicon hydroxyls of the stationary phase for the purpose of improving the peak shape. As a result, TFA was selected as the optimal additive.

In addition, the concentrations of TFA (0.05–0.5%) were also optimized to obtain a more satisfactory peak shape. Figure 4B illustrated that the peak area of target compounds increased with the increase in TFA concentration. However, the peak area of the compound reached its maximum when the concentration was increased to 0.2% (Figure 4B). Meanwhile, the minimum separation degree of 2.84 was higher than that of other concentrations, and the minimum theoretical number of plates of 1680 was also higher than the minimum theoretical number of plates of other concentrations. In conclusion, the concentration of 0.2% TFA was used for further experiments. Moreover, it was found that water could also improve the peak shape and resolution in the pre-experiments. Therefore, water was used as the second additive, and the concentration of water was further optimized. When the water concentration increased from 4% to 12%, the peak response of target analytes continued to increase (Figure 4C). However, the peak shape became worse when the concentration of water was above 8%. In addition, with the increase in the proportion of water, the low-temperature critical fluid was easy to freeze through the stationary phase. Hence, the concentration of 8% water was chosen for the following experiments.

### 3.3. The Selection of Make-Up Solution

When ultrahigh-performance supercritical fluid chromatography was combined with ion mobility mass spectrometry (UHPSFC-IMMS), the samples needed to be injected with make-up solution to protonate the target analytes and enhance MS signal. Therefore, four types of make-up solutions were evaluated, such as formic acid, acetic acid, ammonium formate, and ammonium acetate. As can be seen from Figure 4D, different make-up solutions acting as the X axis and Y axis displayed the peak areas of seven target compounds (Scoparone (1), Dendrobine (2), Apigenin (3), Naringenin (4), Dendrophenol (5), Luteolin (6), and Erianin (7)) detected by UHPSFC-IMMS. The highest peak areas and the best MS signal of target analytes were obtained when acetic acid was used as the make-up solution. The influence of a salt solution was worse than that of an acid solution, probably because of the weak ionic strength. Moreover, the use of salt solutions (ammonium formate and ammonium acetate) would contaminate the diverter back pressure tube and MS. The reason why acetic acid had a better effect than formic acid might be that formic acid was more acidic than acetic acid, which made the hydrogen ions in formic acid difficult to ionize out, thus, reducing the response of the MS signal. The response of formic acid as make-up solution was generally lower than that of acetic acid as the make-up solution. Therefore, acetic acid was chosen as the optimal make-up solution.

### 3.4. The Selection of Instrument Parameters

In addition to the stationary phase, mobile phase, and make-up solution, instrument parameters were another important factor that influenced the separation of target analytes. Instrument parameters included back pressure, column temperature, fragmentor, nebulizer, gas temperature, sheath gas temperature, etc. The SFC conditions were optimized, as shown in Figure 5, and the mass spectrometry parameters are shown in Figure 6. The back pressure of the back pressure regulator varied between 100 and 160 bar and was evaluated under the above optimal conditions, as shown in Figure 5b. In SFC analysis, the main role of back pressure was to control the supercritical state of CO_2_ during the whole process. The density of supercritical CO_2_ improved with increasing back pressure, because CO_2_ was a kind of compressible fluid. Supercritical CO_2_ fluid possessed greater elution and solvating abilities under the higher back pressure. As shown in Figure 5a, with the back pressure increasing, separation selectivity remained nearly unchanged, and the peak area slightly increased. However, due to the higher polarity of the mobile phase, the retention time decreased slightly with increasing pressure. The response to Scoparone (1) and dendrobiol became worse. Therefore, a working value of 115 bar was selected for this BPR pressure. At this point, the separation and theoretical plate number were better. The minimum separation degree was 1.67, which was greater than the minimum separation degree under other back pressures. SFC separations of target analytes have been run at various column temperatures ranging from 20 °C to 80 °C. As seen in Figure 5b, the best extraction effect of Scoparone (1); Dendrobine (2); and Apigenin (3) was achieved when the column temperature increased to 60 °C. Obviously, the column temperature had little effect on the separation and response of Naringenin (4), Dendrophenol (5), Luteolin (6), and Erianin (7). The reason may be that the concentration of modifier (methanol) in the mobile phase was large (more than 15%), resulting in the limited influence in the mobile phase density caused by the column temperature. Therefore, to compromise between the extraction effect and resolution, the better column temperature of 60 °C was adopted for further study.

The parameters of the mass spectrometry that may affect the separation effect have been optimized, including the fragmentor voltage, the drying gas temperature, the drift gas temperature, and the nebulizer gas pressure. The optimization results are shown in Figure 6. The mass spectrometry collection of target compounds have been run at different fragmentor voltages ranging from 340 to 400 V. At this time, the minimum separation was 1.16, which was greater than that of other fragment volumes. Comprehensively, the target compounds showed the highest total intensity at 370 V. The MS parameters had little effect on the number of adducts but mainly affected the strength (ionic strength) of Scoparone (1) and Dendrobine (2). The ionic strength of the target compounds decreased slightly with the increase in drift gas temperature and nebulizer gas pressure. Therefore, 250 °C and 35 psi were the best conditions for the drying gas temperature and spray nebulizer gas pressure, respectively. In addition to the above three parameters, drift gas temperature was another important factor affecting mass spectrum response. Under the same conditions, the variation of temperature between 320 and 380 °C was studied. Thus, in terms of abundance, 380 °C was selected as the optimum drift gas temperature for the next experiments. The number of theoretical plates of each compound was generally higher than that of the corresponding compound under other drift gas temperature, with the maximum value of 14,497.

### 3.5. Determination of Collisional Cross-Sections (CCS)

In addition to retention time and *m*/*z*, IM-QTOF-MS analysis also provided CCS. CCS was a parameter related to molecular morphology and size, so it can supplement accurate mass and fragment morphology. This method effectively provided an additional information for the identification of the isomers of the target compound. By applying calibration, CCS value can be derived from the drift time of each target compound. The single-field CCS measurement was to determine the CCS of the unknown compound based on the CCS value of the known compound under a fixed voltage. The inlet voltage of the fixed drift tube implemented in this experiment was 1700 V, and the calculation formula of CCS (Ω) is as follows:(1)$${\;t}_{D}{=\beta \gamma \Omega +t}_{\mathrm{fix}}$$
(2)$$\gamma =\frac{1}{Ζ}\sqrt{\frac{m_{1}}{m_{1\;}{+m}_{B}}}$$
where Ω is the CCS value of the compound (Å2), β is the correction factor, t_fix_ is the fixed time of the detector, z is the number of charges, m_1_ is the mass of the drift gas, and m_B_ is the mass of the target compound. The multi-field method needed to set eight time period voltages, and the drift voltages were 1000, 1100, 1200, 1300, 1400, 1500, 1600, and 1700 V in sequence. According to the Mason-Schamp equation to determine the CCS of the compound directly and accurately, the calculation formula is as follows:(3)$$\Omega =\frac{{\left(18\pi \right)}^{1/2}}{16}\frac{\mathrm{ze}}{{\left(k_{b}T\right)}^{1/2}}{\left[\frac{1}{m_{1}}+\frac{1}{m_{B}}\right]}^{1∕2}\frac{t_{d}E}{L}\frac{760}{P}\frac{T}{273.2}\frac{1}{N}$$
where E is the electric field intensity, L is the length of the drift tube (0.781 m), *T* is the temperature of the drift gas (Kelvin), P is the pressure of the drift gas, k_b_ is the Boltzman constant (1.3806 × 10 ^−23^ m^2^ kg s^−2^ K^−1^ ), e is the basic charge (1.602 × 10^−16^ C), m_1_ is the mass of the drift gas, m_B_ is the mass of the target compound, z is the number of charges, and N is the drift gas number density at 1 atm and 273 K (2.687 × 10^25^ m^−3^). Figure 7 shows the drift time spectrum of the target ion. Figure 8 shows drift time spectrum of Erianin (7). Table 1 shows the identified compounds, their drift time, measured *m*/*z* values, and CCS values for single and multiple fields. In general, CCS values obtained under multi-field conditions were used as references. The multi-field CCS values of Scoparone (1), Dendrobine (2), Apigenin (3), Naringenin (4), Dendrophenol (5), Luteolin (6), and Erianin (7) were 139.3, 158.3, 158.7, 165.4, 162.4, 163.0, and 170.3, respectively. The relative error of single-field and multi-field was less than 1.41%, which could be used as a reference.

### 3.6. Method Validation

In order to verify the performance of the MSPD-SFC-IM-QTOF-MS method, the evaluation parameters of this method include analysis curve, calibration range, detection limit (LOD), limit of quantification (LOQ), and precision (intra-day and inter-day precision). Table 2 summarizes the evaluation results of the analytical methods (MSPD and SFC-IM-QTOF-MS used in this study. The calibration curve constructed for each target compound has good instrument linearity, and the regression coefficient (R^2^) exceeds 0.991, indicating satisfactory linearity for all the test compounds. The intra-day and inter-day repeatability was expressed by relative standard deviation (RSD), ranging from 0.28% to 9.8%. The intra-day and inter-day repeatability of all analytes were less than 9.8%. According to the AOAC calculation, the results showed that the analysis method had sufficient accuracy. LODs were specified as LOD = 3.3 c/m (where c is the concentration of the standard reference substance and m is the signal-to-noise ratio of the instrument) in the range of 1.4–70 ng/mL. Meanwhile, LOQs were specified as LOQ = 10 c/m and resulted in the range of 4.7–241 ng/mL. These results showed that the developed MSPD extraction approach was a reliable technique for SFC-IM-QTOF-MS determination of seven polar standard compounds from caulis dendrobii.

### 3.7. Analysis of Real Samples

In order to further study the practicability of the established method in the analysis of actual caulis dendrobii samples, the MSPD combined with the SFC-IM-QTOF-MS method was used to extract three real samples. Three samples of Dendrobium Drumstick (Yunnan, China) and Caulis dendrobii (Yueqing, Zhejiang and Yandang Mountain, Zhejiang, China) were purchased from different regions and analyzed by optimized methods. The content results are summarized in Table 3. Chromatogram of Dendrobium sample is shown in Figure 9. Each sample was repeated twice.

In the process of determination, only Dendrobine (2) and Scoparolide (1) could be detected in Dendrobium thunbergii from Yunnan and Caulis dendrobiifrom Yueqing, Zhejiang, while all seven compounds could be detected in Caulis dendrobiifrom Yandangshan, Zhejiang. Therefore, aiming to verify the reliability and applicability of this method, the Caulis dendrobiiin Yandang Mountain of Zhejiang Province was spiked at the concentrations of 0.05 and 1 μg/mL. As shown in Table 3, the average recovery rate of most compounds was between 60.7% and 127%, and the reproducibility RSD of the method was ≤8.4%. The results showed that the method was simple, accurate, sensitive, and suitable for the simultaneous determination of multiple active ingredients in Dendrobium candidum. In order to evaluate the matrix effect (ME), the matrix-matched calibration curves and the solvent-only calibration curves at the same serial concentrations were used. The ME was assessed as expressed by the following formula Equations:(4)$$\mathrm{ME}\left(\%\right)=\frac{\mathrm{peak}\;\mathrm{area}\left(\mathrm{matrix}\;\mathrm{standard}\right)}{\mathrm{peak}\;\mathrm{area}\left(\mathrm{solvent}\;\mathrm{standard}\right)}\times 100$$
and in our study, the matrix effect was between 85% and 96%.

## 4. Conclusions

A rapid and effective SFC-IM-QTOF-MS method was developed for the separation and qualitative analysis of seven major polar compounds in Dendrobium officinale. SFC-IM-QTOF-MS was very important for the analysis of active components in complex matrix. After optimization, all the target compounds were well separated on ZORBAX RX-SIL column with gradient elution within eight minutes. Compared with the traditional HPLC method, SFC method has the advantages of short analysis time, less organic solvent, and environmental protection. The seven analytes were successfully separated within eight minutes. In addition, the quantitative method was established and verified. The results showed that the method had good repeatability and recovery. Due to the need to better characterize complex target compounds, IM/MS analysis was used to evaluate the CCS value of the target analyte at the same level of concentration. By measuring the CCS value of the compound, the accuracy of identifying the target compound was improved. This method has high separation efficiency and provides a new method for the separation and quantification of Dendrobium candidum.

## Figures and Tables

**Figure 1 foods-11-00111-f001:**
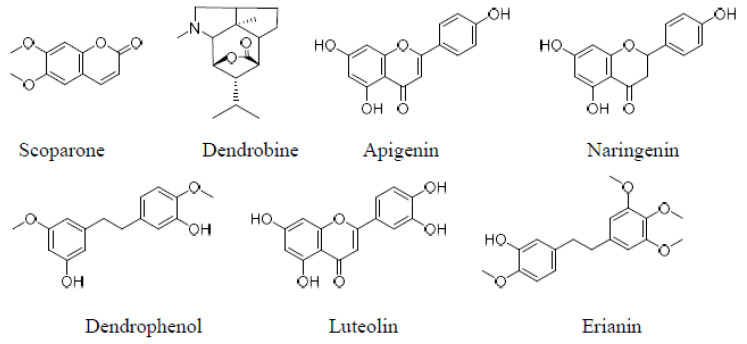
Structure formula of target compounds.

**Figure 2 foods-11-00111-f002:**
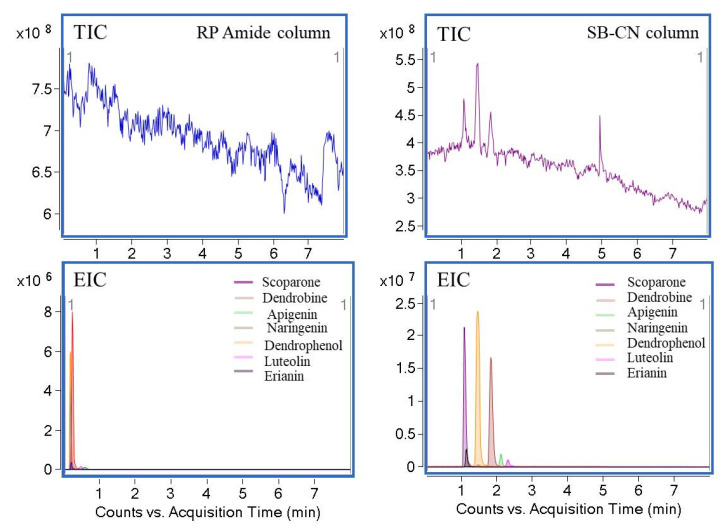
Effect of the RP amide column and Agilent SB-CN column on the separation effect in positive ion hydrogenation mode.

**Figure 3 foods-11-00111-f003:**
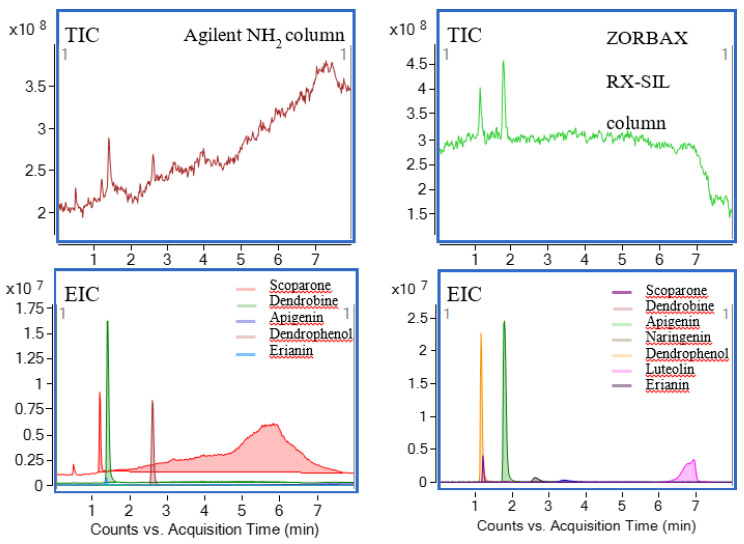
Effect of the Agilent NH_2_ column and ZORBAX RX-SIL column on the separation effect in positive ion hydrogenation mode.

**Figure 4 foods-11-00111-f004:**
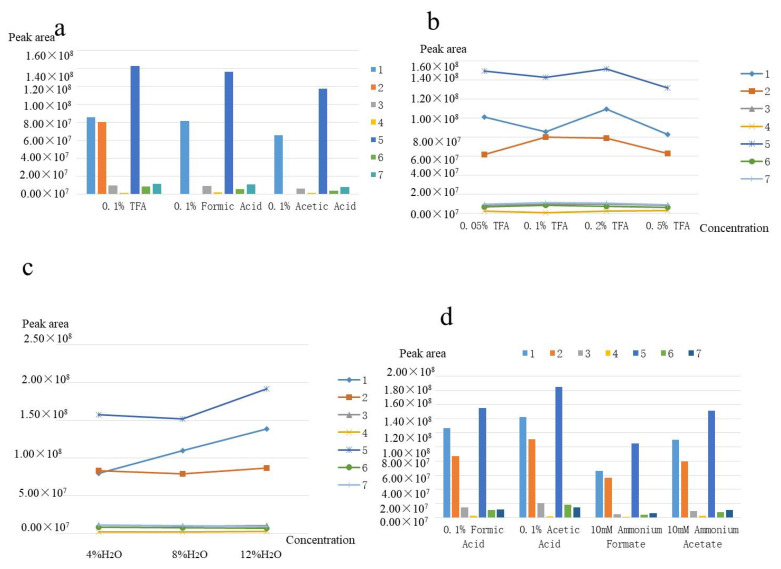
(**a**) Effects of mobile phase additives. (**b**) Effects of additive concentration. (**c**) Effects of water concentration. (**d**) Effects of make-up solution. Analytes: 1. Scoparone (1); 2. Dendrobine (2); 3. Apigenin (3); 4. Naringenin (4); 5. Dendrophenol (5); 6. Luteolin (6); and 7. Erianin (7).

**Figure 5 foods-11-00111-f005:**
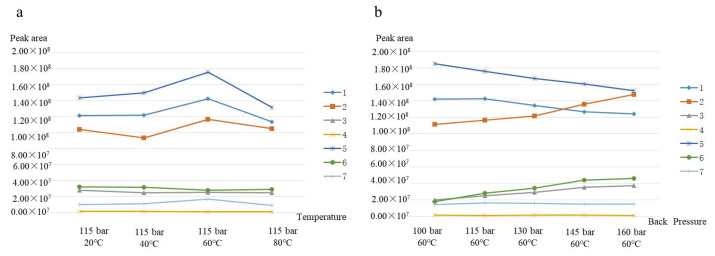
The effect of temperature (**a**) and back pressure (**b**) on the mass spectrum response. Analytes: 1. Scoparone (1); 2. Dendrobine (2); 3. Apigenin (3); 4. Naringenin (4); 5. Dendrophenol (5); 6. Luteolin (6); and 7. Erianin (7).

**Figure 6 foods-11-00111-f006:**
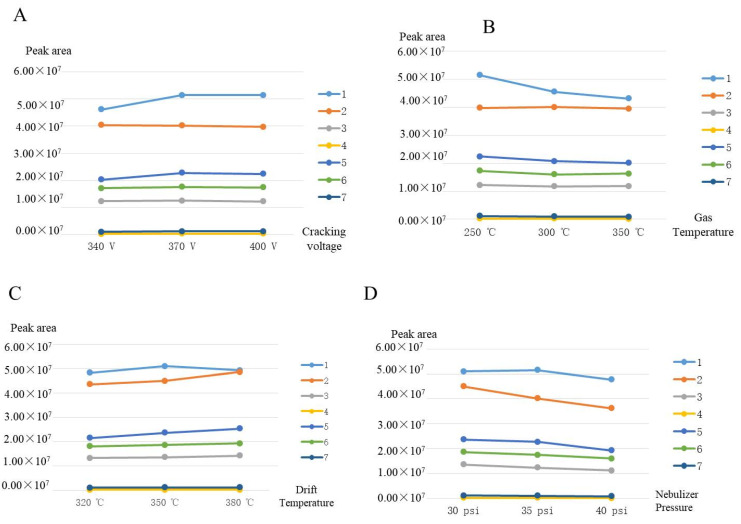
(**A**) Effect of cracking voltage on separation. (**B**) Effect of gas temperature on separation. (**C**) Effects of drift temperature on separation. (**D**) Effect of nebulizer pressure on separation. Analytes: 1. Scoparone (1); 2. Dendrobine (2); 3. Apigenin (3); 4. Naringenin (4); 5. Dendrophenol (5); 6. Luteolin (6); and 7. Erianin (7).

**Figure 7 foods-11-00111-f007:**
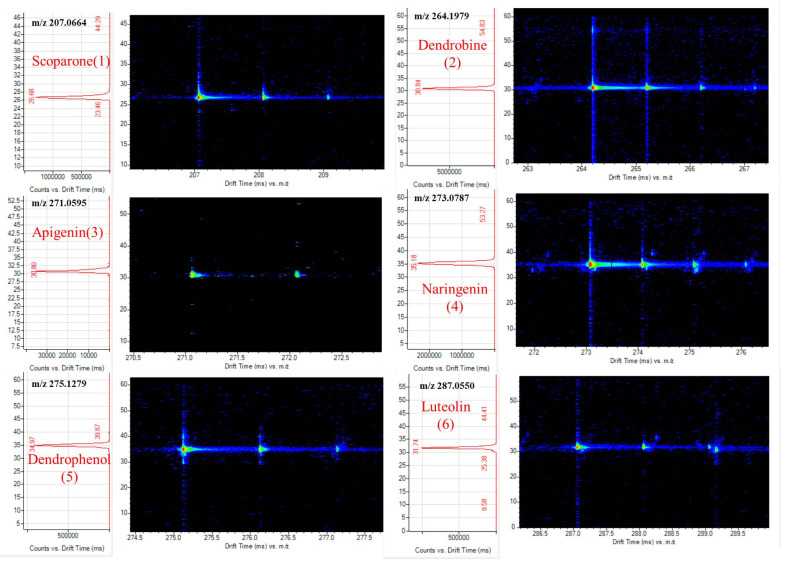
Drift time spectrum of target compounds.

**Figure 8 foods-11-00111-f008:**
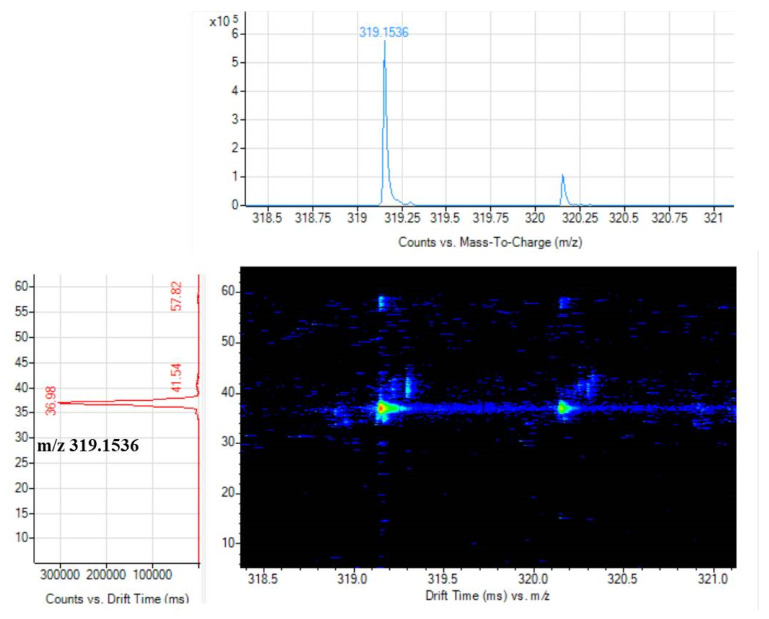
Drift time spectrum of Erianin (7).

**Figure 9 foods-11-00111-f009:**
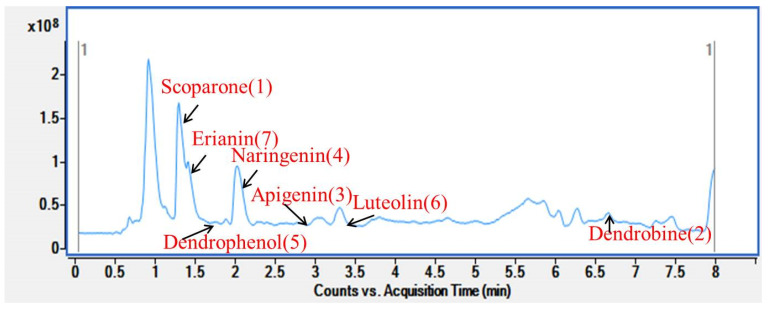
Chromatogram of Dendrobium sample.

**Table 1 foods-11-00111-t001:** Single field and multi field CCS of analytes.

Compounds	[M + H]^+^	Drift Time (ms)	CCS (Single-Field)	CCS (Multi-Field)	RSD (%)	t_0_ (ms)	k_0_ (cm^2^/V·s)
Scoparone (1)	207.065	17.06	137.5	139.3	0.65	4.05	1.539
Dendrobine (2)	264.196	19.76	157.1	158.3	0.38	4.79	1.336
Apigenin (3)	271.06	19.7	156.4	158.7	0.73	4.69	1.331
Naringenin (4)	273.076	20.27	160.8	165.4	1.41	4.64	1.275
Dendrophenol (5)	275.128	20.13	159.6	162.4	0.87	4.75	1.301
Luteolin (6)	287.055	20.29	160.6	163	0.74	4.81	1.29
Erianin (7)	319.154	21.32	167.9	170.3	0.71	5.06	1.23

**Table 2 foods-11-00111-t002:** Linear Regression Data, Precision, Limits of detection (LODs), and Limits of quantification (LOQs) of the Investigated Compounds.

Analytes	Linear Curve			Linear	Precision (RSD%)					
	R^2^	Slope ^a^	Intercept ^a^	Range	Intra-Day (n = 6)		Inter-Day (n = 6)		LODs	LOQs
				µg/mL	Retention	Peak	Retention	Peak	ng/mL	ng/mL
					Time	Area	Time	Area		
Scoparone (1)	0.9964	8,632,000 ± 172,640	159,854.91 ± 3197.0982	0.05–2	0.47	2.1	0.53	4.4	1.4	4.7
Dendrobine (2)	0.9915	62,573,135.75 ± 1,251,462.715	6,537,478.47 ± 130,749.5694	0.05–5	0.38	5.5	0.13	9.2	3.2	11
Apigenin (3)	0.9998	1,991,464.4 ± 39,829.288	−12,502.57 ± 250.0514	0.5–5	0.28	3.7	0.22	8.1	42	139
Naringenin (4)	0.9915	102,001.26 ± 2040.0252	−113,627.48 ± 2272.5496	0.5–2	0.41	2.8	0.72	7.4	70	241
Dendrophenol (5)	0.9969	38,932,354.62 ± 778,647.0924	1,062,015.87 ± 21,240.3174	0.05–1	0.31	4.6	0.36	5.5	2.8	9.8
Luteolin (6)	0.9984	5,679,308.93 ± 113,586.1786	28,224.77 ± 564.4954	0.05–5	0.32	11	0.22	7.8	4.6	15
Erianin (7)	0.9942	7,371,156.15 ± 147,423.123	134,607.59 ± 2692.1518	0.05–1	0.31	2.3	0.54	9.8	8.8	29

^a^ Interval at level of confidence 95%.

**Table 3 foods-11-00111-t003:** The content, average recovery, and reproducibility of the sample were determined.

Analytes	Dendrobium		Added	Recovery %	Reproducibility (RSD%)	
	Officinale					
	µg/mL	mg/g	µg/mL		Peak Area	Retention Time
Scoparone (1)	0.029	0.19	0.05	103	0.019	2.3
			1	127		
Dendrobine (2)	0.091	0.61	0.05	86.5	0.045	3.3
			1	46.4		
Apigenin (3)	0.797	5.32	0.05	101	0.016	2.1
			1	68.2		
Naringenin (4)	1.884	12.56	0.05	95	0.055	6.8
			1	42.3		
Dendrophenol (5)	0.019	0.13	0.05	60.7	0.014	4.4
			1	41.2		
Luteolin (6)	0.988	6.59	0.05	97.1	0.034	8.4
			1	23.8		
Erianin (7)	9.026	60.18	0.05	89.8	0.013	4.6
			1	52.4		

## Data Availability

The data that support the findings of this study are available from the corresponding author upon reasonable request.

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
