# Peer review of "A Rapid and Sensitive Method for the Simultaneous Determination of Multipolar Compounds in Plant Tea by Supercritical Fluid Chromatography Coupled to Ion Mobility Quadrupole Time-of-Flight Mass Spectrometry"

_foods, 2022, doi:10.3390/foods11010111_

Round 1
Reviewer 1 Report
The authors optimized an effective method for the simultaneous determination of seven multipolar compound in plant tea by using Supercritical fluid chromatography coupled to ion mobility QTOF MS.
The method exploit the possibility to have a rapid, effective and green analytical separation by SFC and valuable information on both the structure and accurate mass by the IM QTOF MS technique.
In the paper also the extraction procedure by MSD is optimized.
The paper is well organized and the authors report an impressive number of experiment to optimize all the extraction, analytical and detection parameters which affect the process in order to develop a quantitative method which has revealed as a reliable, rapide and effective method to determine and quantify the seven compounds under investigation in real sample.
I think the paper is clear, relevant for the field and presented in a well-structured manner.
I have some minor comment/correction reported in the following:
- line 19 Abstract “The limits of detection were 1.415~8.824 ng/mL, and the limits of quantification were 4.717~29.412 ng/mL.” I think it would be better to specify at which compound refer the minimum and maximum LOD and LOQ
- Line 26 The correct definition of Supercritical fluid chromatography (SFC) is “a separation using a supercritical fluid as mobile phase”. I know that CO2 is absolutely the most used supercritical fluid currently used but I recommend to specify that, for example by changing in the following statement “is” with “can be” or “generally is” or something like that.
“Supercritical fluid chromatography (SFC) is a chromatography technique in which
- Line 41, 44 and in almost all manuscript in “IM-QTOF-MS” acronym “T” is missing
- Line 44 please rephrase as follow:" IM-QTOF MS combines the characteristics of ion mobility technology, which is sensitive, fast and can provide the information of ion structure, while the QTOF mass spectrometric analyzer can provide the accurate information of mass [10]".
Mass spectrometry, in fact is the overall technique and to be precise (and peraphs a little fussy…) include both an ionic source and an analyser so I think it is not completely correct and too generic to write “while mass spectrometry can provide the accurate information of mass”.
- Line 52 correct the acronym: “supercritical fluid chromatography tandem mass spectrometry (SFC-MS/MS)”
- Line 75 please add to: " Zhang et al. used high performance liquid chromatography coupled to mass spectrometry (HPLC/MS) to analyze trans…."
- Line 88, line 98 and Figure 1: There is a difference between the listed name of the seven standard and those reported in the figure 1, please correct: “scoparone, dendrobine, dendrobiol (is the same of dendrophenol?), luteolin, naringenin, apigenin and magnolia ( is magnolia or magnolian? Sinonymum of erianin???).
- Line 176 Figure 4 caption: please add description of the four graphs (a,b,c,d) and modify on figure b and c the x axis tag and in all graphs the y axis tag (peak area, counts? TIC?).
- Figure 5 Add y axis tag
- Figure 6 caption add description of the four graphs and x and y axis tag on figures
- Line 243: I don’t understand the meaning of “the strength of scoparone and dendrobine” (are the authors referring to the ion intensity??, if yes it would better to write ionic strength or intensity) in the following phrase:
“MS parameters had little effect on the number of adducts, but mainly affected the strength of scoparone and dendrobine.”
By the way it will be useful for the reader to refer in the text to the single compounds also with the number used in the figures.
- Line 264 and 271 probably the formulae must be centered in the text
- Table 1and 2: some words and numbers wrap on the next line, please correct! In table 2 scoparone is bold, why?
- Please move table 2 after paragraph 3.6 method validation. Please add a more explicative caption to table 2 and express LOD and LOQ with the related error (if possible).
- Figure 7 and 8: in the figure m/z is underlined probably the word software correction?
Besides m/z I suggest to report in the figure and/or in the caption also the name of reference compounds and the number used in the other figures
- Figure 8 the mass spectrum image has a very low quality, please improve
- Table 3 please align well rows and columns
Author Response
please see word uploded

Reviewer 2 Report
In the article by Zi-Xuan Yue & Jun Cao, a new method based on supercritical fluid chromatography coupled to ion mobility quadrupole time-of-flight mass spectrometry was employed for the analysis of 7 polar compounds (scoparone, dendrobine, dendrobiol, luteolin, naringenin, apigenin and magnolian.) in plants. For the extraction, a matrix-solid phase dispersion protocol was used. Different factors affecting separation (type of stationary phase, organic modifier, the content of acid additive, pressure and temperature) were optimized using a univariate procedure. The research could be of interest to readers of Foods. However, some issues should be modified to achieve the quality standards of the journal. First of all, the authors employed the peak areas as the reference parameter for the optimization of conditions. This is not the ideal chromatographic factor for this kind of study (see comments below). Moreover, there are some validation problems (for example LOQs are higher than the linear range). Finally, the extraction protocol seems not to be suitable for these analytes. For these reasons, I recommend a major revision. The main lack points are described below:
-Figure 1: Please revise the text because the names indicated in the introduction are not the same that appear in the figure
-Decimal places: Please revise these numbers because 3 decimal places at the ng/mL level are too much.
-Section 2.3: Can the authors indicate the source of the extraction process? Did they optimize the protocol? In that case, the optimization process should be included in the manuscript
-Line 126: Please indicate the temperature and time of the drying process.
-Section 3.1: Please, indicate the rest of the conditions employed during the evaluation of the different stationary phases.
-Sections 3.1, 3.2, 3.4: As can be deduced from the figures, the authors used the peak areas as the decisive factor for chromatographic optimization. Although this parameter can be considered, a solid chromatographic study must contemplate other parameters: number of theoretical plates, the height of the theoretical plate, retention times, peak width, capacity factor, resolution, asymmetry factor, tail factor and selectivity factor (among other).
-Figures: The figure captions should be improved. The captions should be auto explanatory (the text should not be necessary to understand them). For example: indicate the selected ions for the EIC in Fig. 2, indicate the meaning of a,b, c and d for Fig 4 and 6, add additional information in Fig. 6 (correspondence of 1-7, etc.) …
-Table 2 and Section 3.6: there are several problems with validation studies.
- The number of levels of concentrations employed for calibration study should be indicated
- Confidence intervals of slope and intercepts must be included
- LOQs for Dendrobine (10.58 ng/mL), Apigenin (138.89 ng/mL9, Naringenin (2415.46 ng/mL), Luteolin (15.38 ng/mL), and Erianin (29.41 ng/mL) are higher than the starting point of linear range (10 ng/mL) or even than the final point (2000 ng/mL for naringenin). Considering that any concentration lower than the LOQ can not be quantified with confidence, this is a very important point that must be solved.
- Did the authors check the matrix effect? Considering the complexity of the sample and the combined use of chromatography and MS, a strong matrix effect is expected
-Section 3.7:
- Please indicate how many samples were employed for the recovery and reproducibility studies. Once more, the concentrations employed should be considered taking into account the LOQs
- The recoveries indicated in line 325 do not coincide with those in table 3. In fact, approximately half of these values are lower than 60% (which is a very low value for an efficient extraction protocol). This probably could be explained if the extraction method was not optimized by the authors (a reference method only can be employed if it is tested for their particular combination analytes-samples)
-Finally, a chromatogram for a real sample should be included.
Author Response
please see word uploaded

Reviewer 3 Report
Suitable parameters and conditions for SFC separation and MS detection and matrix solid phase dispersion (MSPD) microextraction are investigated for achieved analysis of target analytes, and method validation also has been provided. This developed SFC-MS method can be used for analysis of real samples with rapid analysis. However, minor revision should be clarified as follows
(1) According to Lines 302 and 303 stated that “These results show that the analysis method was sufficiently accurate”, what are the criteria for sufficiently accurate? Typically accuracy of quantitative analysis may be evaluated from %recovery, while precision of quantitative analysis may be evaluated from %RSD. For example, according to the Horwizth equation and HORRAT values, acceptable %recovery and %RSD values depend on the concentration used for testing. AOAC may be used for consideration of acceptable accuracy and precision.
(2) Significant figures for data reports should be considered. Uncertain data, such as %RSD, LOD and LOQ in Tables 2, and %RSD in Table 3, should be reported with one or two significant figures. Two significant figures are suggested. In your case for LOD and LOQ with more than 100, inter numbers may be reported such as 139 or 140 for 138.889, 725 or 720 for 724.638, 2415 or 2400 for 2415.45. In addition, %Recovery with three significant figures in Table 3 should be reported. The data in text should be changed according to the data in Table changed.
(3) Figs 4 to 6 are line or bar graphs showing different analytes with light colors. For clear vision for different analytes, dark color should be used for some analyte, and similar color should be avoided. Although different symbols are used, but they are quite small to differentiate.
(4) Typing errors should be corrected. Such as data numbers in line 273, subscripts of symbols in lines 265, 266, and 273. Names of compounds in Tables are not in the same line.
Author Response
please see word uploaded

Round 2
Reviewer 2 Report
Some aspects that were included in the response to reviewers should be indicated in the text. Moreover, certain issues previously indicated were not modified or discussed.
-Regarding the use of chromatographic parameters, the authors included some mentions but did not indicate the numerical values that support their affirmations.
-The selected ions for EICs in figures 2 and 3 were not included
-Please indicate the number of replicates of each sample used during the recovery study
-Regarding the comparison of their method with the extraction with methanol (Pharmacopeia) I couldn´t find the mention of the comparison in line 325 or even in another part of the paper
Author Response
Please see uploaded word
